# Multi-Analytical Analysis of Decorative Color Plasters from the Thracian Tomb near Alexandrovo, Bulgaria

Georgi Avdeev [1], Rositsa Kukeva [2], Denitsa Yancheva [3], Valentin Mihailov [4], Vani Tankova [4], Momtchil Dimitrov [3], Georgi Nekhrizov [5], Radostina Stoyanova [2] and Bistra Stamboliyska [3],*

1    Institute of Physical Chemistry, Bulgarian Academy of Sciences, Acad. G. Bonchev Str., Build. 11, 1113 Sofia, Bulgaria; g_avdeev@ipc.bas.bg
2    Institute of General and Inorganic Chemistry, Bulgarian Academy of Sciences, Acad. G. Bonchev Str., Build. 11, 1113 Sofia, Bulgaria; rositsakukeva@yahoo.com (R.K.); radstoy@svr.igic.bas.bg (R.S.)
3    Institute of Organic Chemistry with Centre of Phytochemistry, Bulgarian Academy of Sciences, Acad. G. Bonchev Str., Build. 9, 1113 Sofia, Bulgaria; denitsa.pantaleeva@orgchm.bas.bg (D.Y.); momtchil.dimitrov@orgchm.bas.bg (M.D.)
4    Institute of Solid State Physics, Bulgarian Academy of Sciences, 72 Tzarigradsko Chaussée, 1784 Sofia, Bulgaria; valentin@issp.bas.bg (V.M.); vtankova@issp.bas.bg (V.T.)
5    National Archaeological Institute with Museum, Bulgarian Academy of Sciences, 2 Saborna Str., 1000 Sofia, Bulgaria; nehrizov@gmail.com
*    Correspondence: bistra.stamboliyska@orgchm.bas.bg

**Abstract:** In the present contribution, we report the results from a study on the ancient technology used to create decorative color plasters in the Thracian tomb near the village of Alexandrovo, Bulgaria. A series of fragments of red, black, grey, white and brown colored lime plasters from the dromos and funeral chamber were investigated by laser-induced breakdown spectroscopy, X-ray diffraction analysis, infrared spectroscopy, paramagnetic electron resonance spectroscopy and differential scanning calorimetry. Based on the combined analytical data, it was possible to identify the pigments, fillers and other materials in the composition of the decorative plasters in the interior, as well as to clarify the technological features related to the plaster creation. The results demonstrated that the murals were implemented on two layers in the case of white, black, grey and brown decoration—first coarse mortar, followed by a white, fine mortar, which usually was made of calcite. In the case of red decoration, a pigment was added to the fine mortar to achieve a colored surface. The pigments were identified as mostly traditional mineral pigments—calcite, kaolinite, red natural ochres (colored earth), brown colored earth and black pigment (amorphous C). The use of the fresco technique is implied by the major participation of calcite and the absence of organic binder in all of the painting layers.

**Keywords:** Thracian tomb; Alexandrovo village; Bulgaria; LIBS; XRD; EPR; IR; DSC

## 1. Introduction

The Thracian civilization reached its peak between the fifth and third centuries BC in present-day Bulgaria. Recent excavations have uncovered a series of tombs containing gold, silver, and bronze artifacts, providing evidence of the high level of culture achieved [1]. The unique tomb at the village of Alexandrovo was discovered accidentally at the end of 2000 in a large burial mound 1 km southeast of the village. During the subsequent rescue excavations, archaeologist Georgi Kitov and his team uncovered its entrance and explored and documented the tomb. The tomb dates back to the end of the fourth century BC [2]. It was found that it consists of a long corridor (dromos), a pre-burial chamber and a burial chamber. It is oriented in an east–west direction, with the entrance facing east. The corridor is made of different sized blocks in irregular rows and is covered with horizontally stacked stone slabs. The ante-burial chamber is quite small. It has a rectangular plan and is covered

by a trapezoidal cantilevered vault. The circular burial chamber has a beehive-shaped dome made of segmental blocks, with the vaulting starting from the first row. It contained a burial bed that was found destroyed (Figure 1).

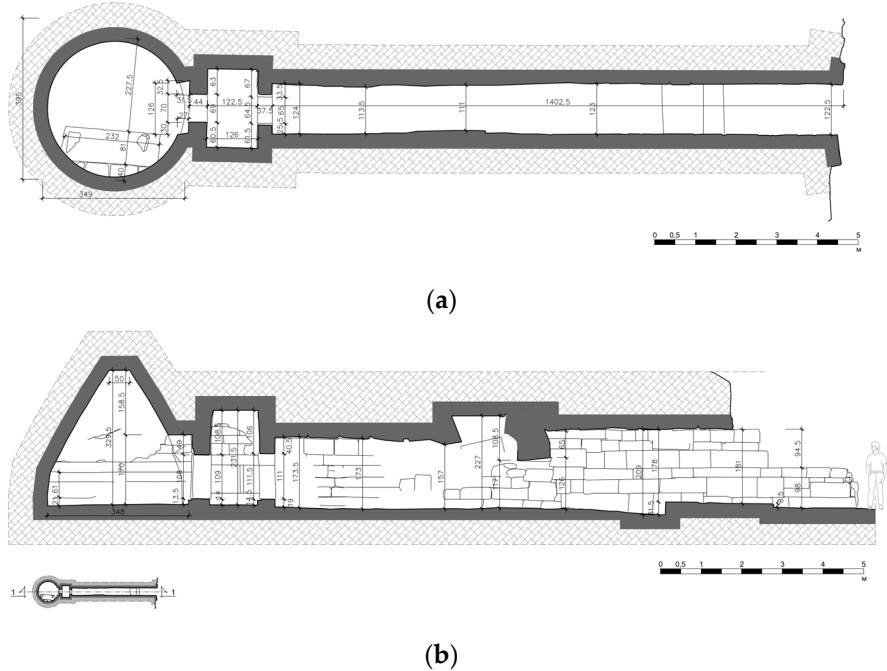

(**a**)

(**b**)

**Figure 1.** Plan 1:50 (**a**) and Section 1:50 (**b**) of the tomb at the village of Alexandrovo (author M. Kamenova, 2021).

The two chambers were constructed of precision-worked blocks with rusticated faces and were closed with double-leaf stone doors. The floors of the three chambers were covered with stone slabs, with a thick mortar applied to the floor in the burial chamber [3]. The tomb was constructed by blocks of rhyolite tuffs, widespread in the Eastern Rhodopes [4].

The walls of the chambers and the westernmost part of the dromos are covered with mortar plaster, on which the frescoed decoration of the tomb is located. It makes this monument one of the most significant representatives of Thracian tomb architecture with artistic decoration. On the north and south walls of the dromos, there are poorly preserved battle scenes between horsemen and foot warriors. The walls of the ante-burial chamber are covered with several parallel, horizontal bands, the first of which is white, the second one consists of black panels separated by vertical white bands, and the third one is ornamented with white spiral motifs on a dark background. The gabled roof is covered with a red-colored plaster and the horizontal part with white. The trapezoidal lintels above the entrances also depict battle scenes, although they have not been fully preserved (Figure 2).

The burial chamber is the most richly decorated. At the base is a deep red-shaped plinth, above which is a figural frieze depicting several human figures on a white background, interpreted as a funeral feast scene. Unfortunately, much of this frieze has been destroyed. Above the frieze follows a strip in black, of which a small part also survives, and above this a broad band colored red. On it, opposite to the entrance of the chamber, an inscription "ΚΟΞΙΜΑCΗC ΧΡΗCTOC" is scratched in two lines and depicts the face of a young man in profile on the left. Experts differ in their opinions about this unique display of individuality. Kitov suggests that it is the signature of the artist who painted the tomb, while other researchers hold the opinion that the graffiti was drawn later and not related to the construction and decoration of the tomb. However, the scholars agree that the name Kojimasis is Thracian [5–8]. The most interesting and best preserved is the frieze, with hunting scenes applied over several decorative bands in the upper part of the dome (Figure 3).

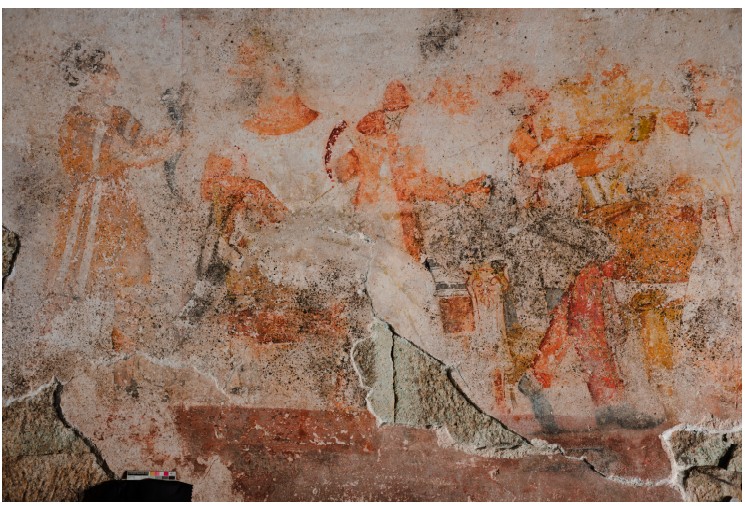

**Figure 2.** Part of the funeral feast scene from the chamber of the tomb at the village of Alexandrovo (author A. Cosentino, 2019).

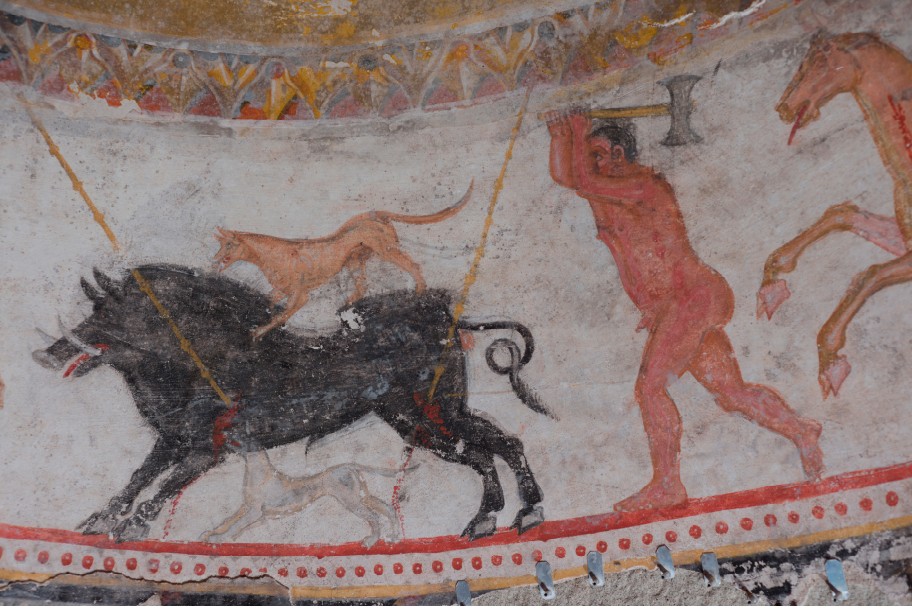

**Figure 3.** Part of the frieze with hunting scenes from the chamber of the tomb at the village of Alexandrovo (author A. Cosentino, 2019).

Understandably, this frieze has aroused the greatest interest among scholars and the general public and gives the fullest picture of the skill of the artists who painted the tomb's interior in the late fourth or early third century BC.

In his publications, the discoverer of the tomb at the village of Alexandrovo, Georgi Kitov, mentions that during its exploration the fallen frescoes were carefully collected and sorted according to the place of discovery. Today the fragments of the frescoes are kept in the Museum Centre for Thracian Art in the Eastern Rhodopes near the village of Alexandrovo. Several of them have been analyzed within the current study.

While the tomb at the village of Alexandrovo is well studied from the historical and artistic point of view [5,9–15], analytical research on the painting materials that were used is limited to date. Some reports concerning the technological characteristics of monochrome decorative wall plasters are available in the literature [16–18]. Detailed knowledge of material composition is essential for art historical or archaeological research and for determining proper conservation and restoration procedures.

The aim of the present study is to identify, with the help of the combined analytical methods, the paint materials used in the decorative colored plasters in the interior of the Thracian tomb near the village of Alexandrovo, Bulgaria, as well as to clarify the technological features related to the making of the plaster. It should be noted that this is the first time that such a complex analytical approach has been applied to pigments and pigmented plasters from Thracian tombs. Thus, the current research will expand our knowledge of Thracian construction skills, as well as the ability of the Thracian elite to acquire, adapt, and further develop the tomb-building technologies known at the time.

## 2. Analytical Techniques Applied in the Study and Related Works

Since a single analytical technique cannot provide complete information about the materials and techniques used in the artwork, usually complementary methods of examination and analysis have to be carried out in parallel. Globally, in recent years, the multidisciplinary approach has been extensively used in the study of the composition of various materials of art and archaeological artifacts [19–21]. Because of the complex nature of the studied materials, in our study, we have adopted a multi-analytical approach to characterize the elemental composition, mineral phases, pigment, binder and filler materials, characteristic impurities and thermal properties of the plasters.

For the analysis of the elemental composition of micro samples, we have chosen Laser-Induced Breakdown Spectroscopy (LIBS). The capacity of LIBS for the identification of pigments in various painted artworks and cultural heritage objects has been demonstrated in many studies. In some studies, LIBS has been utilized for the characterization of pigments in the decorations and the colored layers of archaeological ceramics [22–25] and in the colored glaze of porcelain [26]. In the field of archaeometric analysis of pigments in murals and painted plasters, LIBS has been applied in several studies [27–31].

Paramagnetic electron resonance (EPR) spectroscopy finds application in archaeological science to study the origin of carbonate materials used in ancient marble sculptures [32,33] or as a preparation under layers in wooden picture masterpieces [34]. Investigation of different pigments in objects from cultural heritage could also be assisted by EPR spectroscopy. As many pigments originate from colored transition metal complexes or free radicals, the study of EPR of blue and green pigments [35,36] as well as of black pigments [32,37] can give additional knowledge of their composition or origin. In our study, EPR spectroscopy was used for the identification of the pigments and other materials in the composition of the preparatory layers and the decorative color plasters.

Infrared spectroscopy is a very useful analytical technique for both organic and inorganic materials in diverse archaeological and art contexts—building materials, mortars and mural decoration [38–42], icons, easel paintings and illuminated manuscripts [43–45], textile [46], archaeological and paleontological remains [47]. The attenuated total reflectance Fourier transform infrared (ATR-FTIR) technique offers the additional advantage of easy manipulation, chemically non-destructive analysis with a very small amount of sample or in particular cases—the possibility of non-invasive analysis. Moreover, combining spectral analysis with elemental composition data and dedicated spectral libraries allows the determination of a broad spectrum of materials. In the study of the mural decoration of Alexandrovo's tomb, the ATR-FTIR analysis provided simultaneous information on the presence of particular groups of inorganic materials such as silicate, carbonate, clay materials and organic matter in the studied samples.

The exact crystallographic and mineralogical composition was established based on the X-ray Powder Diffraction (XRD) method. X-ray powder diffraction analysis is widely applied in the diagnostics and conservation of cultural heritage materials to determine mineral compounds, textural properties of the crystalline phases, crystallite size and in-depth distribution [48,49]. Mural paintings from different periods were characterized successfully with XRD methods providing valuable information on mortar, pigment and filler composition [50–54].

The thermoanalytical methods, including thermogravimetry (TG), derivative thermo-gravimetry (DTG), differential thermal analysis (DTA), and differential scanning calorime-try (DSC), enable an analysis of painting media, wood and fossils in archaeological findings, manuscript parchments, mortars and pigments in historic buildings, synthetic polymer coatings, model and historic textiles and tapestries [55–57]. Although destructive in general, the thermal analysis requires only a few milligrams of sample and offers great sensitivity in terms of temperature and weight changes. Recently, it was successfully applied for the identification of organic additives in the mural decoration of Thracian monuments [17,18]. Because of the possibility of the method, in the current study, we have used the technique for the characterization of the thermal properties of the studied plasters and to support the identification of the pigments and the possible presence of an organic binder.

## 3. Materials and Methods

Small fragments (2–5 cm) representing all main colors encountered in the mural decoration of the tomb were collected and studied (Figure 4). Photos with respective scales are provided in the Supplementary Materials (Figures S1–S8). A series of powder samples from the colored plaster layers were obtained by scratching the painted surface of the fragments with a scalpel.

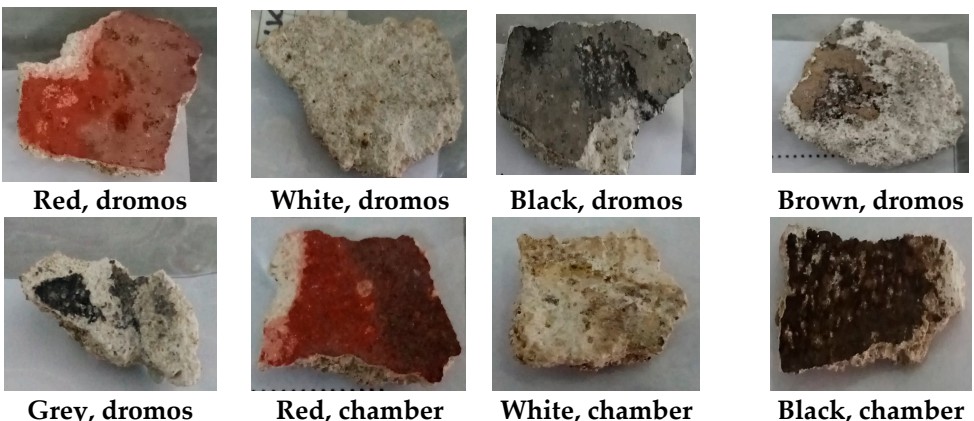

|  |  |  |  |
|---|---|---|---|
| **Red, dromos** | **White, dromos** | **Black, dromos** | **Brown, dromos** |
| **Grey, dromos** | **Red, chamber** | **White, chamber** | **Black, chamber** |

**Figure 4.** Representative color-plastered fragments from the Thracian tomb near the village of Alexandrovo, Bulgaria.

Laser-Induced Breakdown Spectroscopy (LIBS) and X-ray Powder Diffraction (XRD) analysis were carried out directly on the mural fragments. Paramagnetic Electron Reso-nance Spectroscopy (EPR), Attenuated Total Reflectance Fourier Transform Infrared Spec-troscopy (ATR-FTIR) and thermal analysis measurements (DSC and TG analysis) were performed with the powder samples.

The LIBS analysis was performed by a Q-switched Nd:YAG laser. The laser was operated at a fundamental wavelength of 1064 nm with a pulse energy of 8 mJ and a pulse duration of 10 ns. The study was performed by applying 10 consecutive laser shots to the same location. For registration of the spectra, an Echelle spectrometer (Mechelle 5000, Andor Technology, Belfast, UK), with a grating of 52 grooves/mm and inverse linear dispersion of 1–4 nm/mm, was used. All spectra were registered at a delay time of 1 µs after the laser pulse and a gate of 1 µs. The resulting spectra were processed with Solis 64 software. The NIST atomic spectra database was used to identify the observed spectral lines and to determine the elements in the samples. A more detailed description of the experimental setup can be found in our previous work [58].

The EPR analysis was performed on a Bruker EPR spectrometer EMX Premium X working in the X-band at 9.4 GHz.

The materials underwent X-ray diffraction (XRD) examination using a Panalytical Empyrean diffractometer with Highscore Plus software (Scan Axis: Gonio; scan step length of 13.77 s with a continuous scan type; start position 5.0118 °2Th; end position 89.9868 °2Th;

step size 0.013 °2Th). For phase identification, a COD (Crystallography Open Database) database was used [59].

The FTIR and ATR-FTIR spectra were recorded on Bruker Invenio R FT spectrometer at a resolution of 2 cm$^{-1}$ and 100 scans. The samples were measured in solid state (CsI pellet), and for the ATR technique, they were directly applied on a diamond crystal.

The TG-DSC analysis was carried out using a simultaneous thermal analyzer, the STA 449 F3 Jupiter (NETZSCH-Geratebau GmbH, Selb, Germany) apparatus, under the following conditions: temperature interval: 50 °C to 500 °C, temperature heating rate of 10 °C/min and static air atmosphere. Individual samples were placed in platinum pans.

## 4. Results

### 4.1. LIBS Analysis

LIBS analysis, in our case, is limited to the qualitative determination of elements in the investigated fragments due to the lack of suitable reference standards for plasters. However, due to the fact that the intensities of the spectral lines of a given element depend on its concentration in the sample, a semi-quantitative analysis can be made by determining the proportion of spectral line intensities for different samples. Although there are small differences in ablation in different fragments, the ratio of the intensities of the spectral lines is indicative of the relative abundance of the elements. In order to be able to compare the intensities of the spectral lines of the elements, the experimentally measured spectra of the samples were normalized by the total plasma emission intensity method.

LIBS analysis revealed the presence of Ca, Si, Mg, Al, Fe, Na, K, Mn and Sr in all of the samples but with different intensities of the spectral lines. A representative LIBS spectrum obtained for one of the analyzed fragments is demonstrated in Figure 5, and the spectral lines used to identify the elements are listed in Table S1.

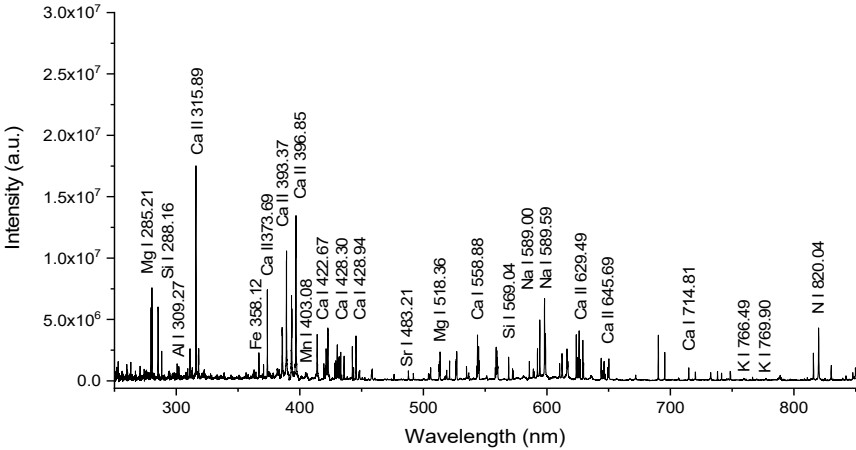

**Figure 5.** Exemplary LIBS spectrum obtained from one of the investigated fragments (red, dromos) representing the spectral range from 250 nm to 850 nm with some of the characteristic spectral lines of the detected elements.

It was noticed that in all of the fragments, the most abundant elements are Ca, Si and Mg. The presence of Ca and Si shows that the fragment material is a mixture mainly of calcium-bearing minerals and silicon-bearing minerals. This is a trivial finding because it is well-known that since ancient times mortar has been made commonly of lime (CaO), sand (SiO$_2$) and water [60]. The detection of strong magnesium spectral lines is not expected and assumes a significant presence of Mg-bearing minerals, most probably dolomite.

Iron is detected in all fragments (Figure 6), but the strongest iron spectral lines are observed in the red painted fragments (red, dromos and red, chamber). In the two black and in the brown painted fragments, the intensity of iron spectral lines decreases. The weakest iron spectral lines are detected in the spectra of white fragments (white, dromos and white, chamber) and the grey fragment (dark grey, dromos).

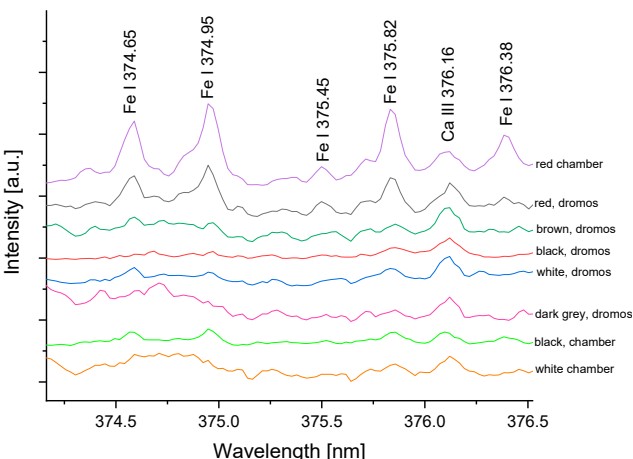

**Figure 6.** Section of LIBS spectrum from 374.5 nm to 376.5 nm showing an increased amount of Fe in colored layers of the fragments painted in red–red, chamber and red, dromos (the spectra are normalized to total plasma emission intensity).

To illustrate the above, a representative section of the spectra of two fragments with the most increased amount of iron oxides (red, chamber) and the one with the least (white, dromos) are compared in Figure S9a in the Supplementary Material. For comparison, iron and calcium spectral lines are designated. This reveals a considerable amount of iron oxides in the red-painted fragments, which most likely is red ochre considering that red ochre was commonly used in antiquity for red paint. It is interesting that in the red-painted fragments, an increased aluminum content compared to the other fragments was observed (Figure S9b), and this indicates the presence of aluminum-bearing minerals.

Kaolinite is known to be a commonly used material in ancient times as a white pigment, but it was also used as filler in paints [61]. Therefore, we can assume that kaolinite may have been added to the analyzed fragments to give a lighter shade to the red paint or was used as filler.

The black pigments used in antiquity are mainly carbon-based pigments in the form of charcoal, soot or graphite, manganese oxides and/or hydroxides and iron oxides [60–62]. In all of the analyzed fragments, manganese was observed only as a minor element. From this, it can be assumed that manganese is not intentionally added but rather naturally present in the raw material; hence, this element has no relation to the coloring of the plaster. Another possibility for the black coloration is carbon, but its detection is a challenging task due to its relatively poor spectrum and the low sensitivity of our analysis in the region of its analytical lines. Therefore, we cannot conclude with certainty whether carbon pigments or iron oxides were used for the black paint.

The detection of potassium and sodium in the spectra of all of the fragments could be attributed to the presence of alkali feldspars, which are one of the main components of mortars [63–65].

The presence of strontium in the mortars is not unusual since this element is naturally found in earth materials. Strontium often substitutes calcium in carbonates, sulfates, feldspars, mica, clays etc. [66–68].

*4.2. EPR Analysis*

The EPR analysis was carried out on a small amount of powder material, scraped from all types of layers building each one of the plasters. Figure 7, left, presents the general view of layers. It was established that most of the samples consist of three layers—coarse mortar, fine mortar and surface pigmented layer, while a "less colored underlayer" exists only in mortars with red color (red, dromos and red, chamber). Two samples deviate from the general appearance. The white sample from the dromos represents a one-layer slab with smoother upper and rougher lower surfaces. The brown sample from the dromos consists of only fine mortar with a pigmented upper part.

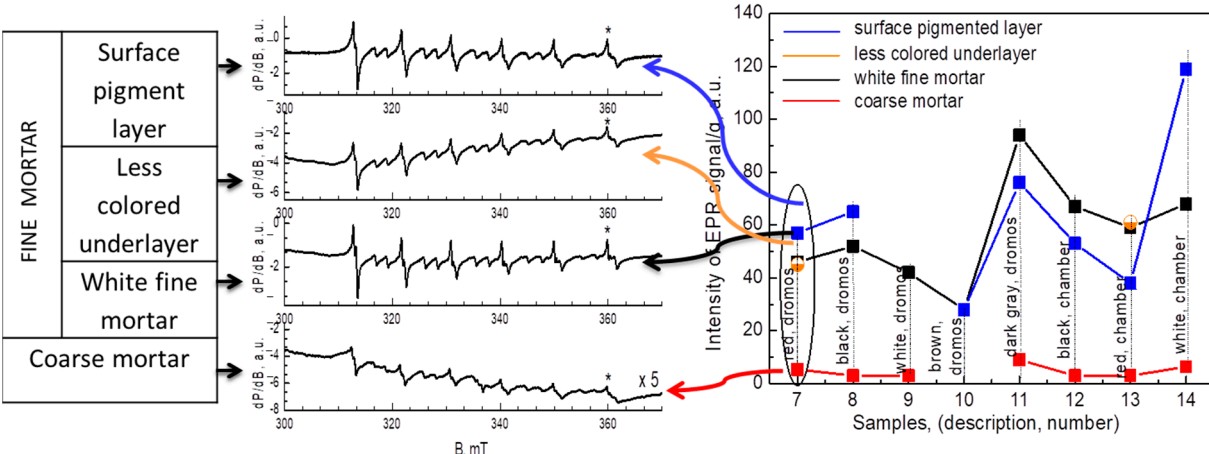

**Figure 7.** EPR analysis of surface-pigmented, fine and coarse mortar material sampled from fragments of the Alexandrovo tomb. Right—intensity of EPR signal of calcite in different layers, middle—EPR spectra of sample red, dromos; left—general view of the layers.

The EPR study has been focused on the investigation of carbonates, which are the main component of lime plasters. Although not paramagnetic, carbonates usually exhibit an EPR signal due to impurities of $Mn^{2+}$ ions, which substitute for $Ca^{2+}$ ions in the crystal lattice of calcite $CaCO_3$ and both for $Ca^{2+}$ and $Mg^{2+}$ ions in dolomite $CaMg(CO_3)_2$. Figure 8 presents the EPR spectra of $Mn^{2+}$ ions in standard samples of calcite and dolomite that have been used as comparative spectra. The EPR analysis reveals that the portion taken from each layer is present only in the form of calcite or dominantly carbonate. To illustrate this assertion, Figure 6 presents spectra of isolated $Mn^{2+}$ ions incorporated in the calcite used for the surface and fine mortar of the sample black, dromos.

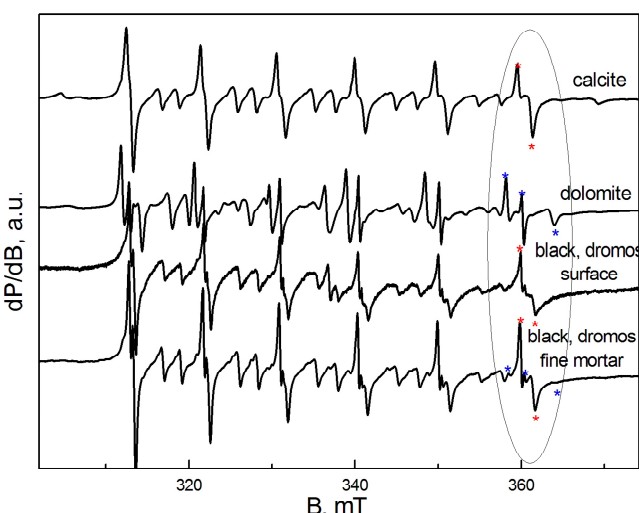

**Figure 8.** EPR spectra of calcite and dolomite standard minerals, the spectra of a pigmented surface and a fine mortar of sample black, dromos are also given.

The EPR spectrum of $Mn^{2+}$ ions allows us not only to determine the chemical form of the carbonates but also to estimate the calcite content in each sample. Figure 7 shows the intensities of the sixth line in the spectrum of isolated ions (marked with *) recalculated for a 1 g sample. As seen, the amount of calcite is lower for the coarse mortar. The surface pigment layer and corresponding underlayer in red-colored samples (red, dromos and red, chamber) display identical calcite content. The last observation allows the conclusion that they were created by the same main material, and a pigment was added to the fine mortar in order to obtain a colored surface.

In the EPR spectra of the samples pigmented in black or brown color, one additional signal is observed between the third and fourth lines of the $Mn^{2+}$ hyperfine structure. Figure 9 presents the spectra of samples black, dromos; brown, dromos; grey, dromos and black, chamber between 329 and 343 mT, as the extra signal not belonging to $Mn^{2+}$ isolated ions is indicated. These signals exhibit g-factors varying from 2.0028 to 2.0031 and a narrow linewidth, $\Delta H_{pp}$, of about 0.33–0.45 mT. The EPR parameters allow the signal to be related to the presence of radical species in the samples. Moreover, the established g-factor values are characteristic of carbon radicals [69], which was confirmed by the EPR analysis of a series of black pigments (Table S2). Previous studies of sources of black pigments, such as cremated bones [70], bituminous coal [71] and Beechwood [72], show very close values of g-factors (Table S2), and the only way to identify them is by using their linewidth values. The peak-to-peak linewidth in the spectra of radicals in Alexandrovo dark/brown colored fragments is in good agreement with the carbon radical signal width observed in materials of plant origin ashed at different temperatures, with the closest values found for beech wood [69]. The possibility of using, as a pigment, cremated animal bones or coal material is unlikely as their linewidth differs significantly from those we found in the studied plasters (Table S2).

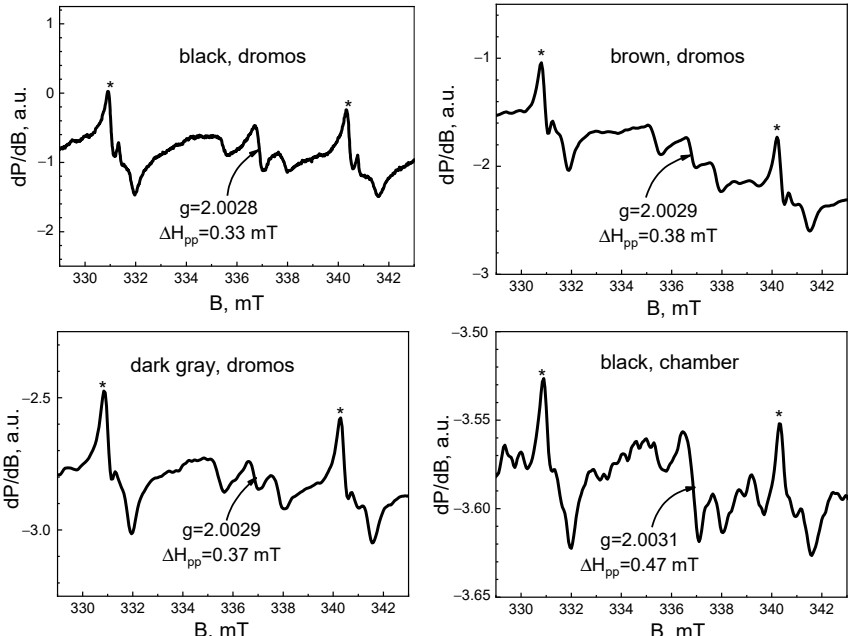

**Figure 9.** EPR spectra of black, brown and dark gray pigments from the Alexandrovo tomb.

### 4.3. X-ray Diffraction Analysis

The components in the pigmented layers and the underlayers lying beneath established by the XRD analysis are listed in Table 1. The XRD patterns are provided in the Supplementary Material (Figures S10–S17).

In both places—dromos and chamber, the painted layer is applied on top of a preparatory layer (fine mortar). The preparatory layer, lying beneath the paint, is mainly composed of lime mortar and, in the case of red layers, hematite and lime. In the lime used to prepare the plaster, there is a certain amount of aragonite, which is usually associated with the presence of fossils in the rock material from which the lime was prepared. In all cases where pigment was added, kaolinite is also present. As mentioned above, kaolinite was used as a white pigment and as a carrier of other pigments for better adhesion.

**Table 1.** Description of studied samples and components in the pigmented layers and the underlayers lying beneath established by XRD analysis.

| Color | Location | Identified Components in the Pigments Layer | Identified Components in under Layer |
|---|---|---|---|
| Red | Dromos | Calcite ($CaCO_3$)<br>Hematite ($Fe_2O_3$)<br>Kaolinite ($Al_2Si_2O_5(OH)_4$) | Calcite ($CaCO_3$)<br>Gypsum ($CaSO_4.2H_2O$)<br>Kaolinite ($Al_2Si_2O_5(OH)_4$)<br>Quartz ($SiO_2$)<br>Hematite ($Fe_2O_3$) |
| Black | Dromos | Calcite ($CaCO_3$)<br>Gypsum ($CaSO_4.2H_2O$)<br>Muscovite ($KAl_2[AlSi_3O_{10}](OH)_2$) | Calcite ($CaCO_3$)<br>Dolomite ($CaMg(CO_3)_2$)<br>Aragonite ($CaCO_3$) |
| White | Dromos | Calcite ($CaCO_3$)<br>Dolomite ($CaMg(CO_3)_2$) | |
| Grey | Dromos | Calcite ($CaCO_3$)<br>Apatite ($Ca_5(PO_4)_3(F,Cl,OH)_2$) | Calcite ($CaCO_3$)<br>Quartz ($SiO_2$)<br>Bunsenite (NiO) |
| Brown | Dromos | Calcite ($CaCO_3$)<br>Muscovite ($KAl_2[AlSi_3O_{10}](OH)_2$)<br>Albite ($NaAlSi_3O_8$) | Calcite ($CaCO_3$) |
| Red | Chamber | Calcite ($CaCO_3$)<br>Hematite ($Fe_2O_3$)<br>Kaolinite ($Al_2Si_2O_5(OH)_4$)<br>Aragonite ($CaCO_3$) | Calcite ($CaCO_3$)<br>Aragonite ($CaCO_3$)<br>Hematite ($Fe_2O_3$) |
| White | Chamber | Kaolinite ($Al_2Si_2O_5(OH)_4$)<br>Gypsum ($CaSO_4.2H_2O$)<br>Quartz ($SiO_2$) | |
| Black | Chamber | Graphite (C)<br>Muscovite ($KAl_2[AlSi_3O_{10}](OH)_2$)<br>Kaolinite ($Al_2Si_2O_5(OH)_4$)<br>Gypsum ($CaSO_4.2H_2O$) | |

The appearance of calcite in the diffraction patterns is explained by its presence in the plaster, which is a lime plaster and a thin layer on the surface of the frescoes formed over years as deposits from the groundwater. The presence of muscovite is attributed to soil contamination on the surface of the studied fragments. Quartz and albite are mineral components originating from the sand used in the coarse mortar.

The study of the phase composition of the pigments gives an idea of their mineral composition: white—kaolinite; black—graphite; red—hematite; dark grey—mixture of kaolinite with black pigment. In the latter case, the XRD analysis has indicated apatite presence, which is usually related to the use of bone black pigment. However, as the EPR analysis has shown that the carbon radicals in the dark grey fragment reveal characteristics closer to a carbon black pigment of plant origin, it should be considered that most probably the bone black pigment was added in smaller amounts or the apatite is present as a natural admixture of the white pigments. The following ICSD (Inorganic Crystal Structure Database) reference cards were used for phase identification: Calcite 98-015-8257; Aragonite 98-001-5194; Hematite 98-008-2135; Kaolinite 98-003-0285; Gypsum 98-002-7221; Quartz low 98-009-0145; Muscovite 98-002-6818; Albite 98-007-7421; Apatite 98-009-9362; Graphite 98-003-1829.

### 4.4. IR Analysis

The powder samples were examined by ATR Infrared spectroscopy to provide information on the presence of different inorganic and organic materials and, in this way, to identify the painting technique. The characteristic bands for the materials identified by the

ATR-FTIR analysis are summarized in Table S3. The analysis showed the content of a large amount of calcium carbonate in all tested samples. On the other hand, none of the spectra of the studied samples showed any organic matter.

The ATR-FTIR spectra of the powder samples collected from the white-colored plasters from dromos and chamber are shown in Figure 10. Both of these spectra evidenced the characteristic peaks of calcite at 1398–1407, 873, and 712 cm$^{-1}$ attributed to the asymmetric stretching and out-of-plane and in-plane bending of $CO_3^{2-}$ ions, respectively [73,74]. A weak band is observed at 2517 cm$^{-1}$ also due to calcite [75]. The position of the bands at 872 and 712 cm$^{-1}$ indicated that the samples contain mainly the calcite form of calcium carbonate and not the other polymorphic forms of aragonite or vaterite [76].

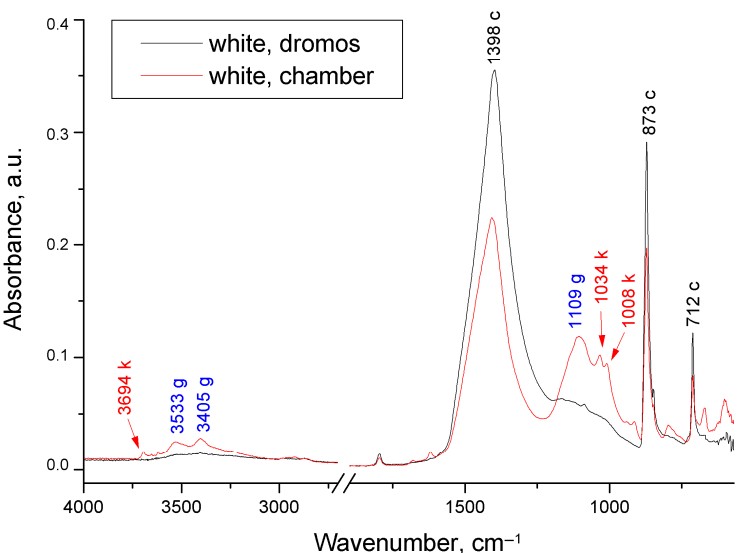

**Figure 10.** ATR-FTIR spectra of powder samples collected from the white-colored surface layers of dromos and chamber; the characteristic bands of identified materials are denoted by letters: c (calcite), k (kaolinite) and g (gypsum).

The analysis revealed that the powder sample collected from the white surface layer of the dromos solely contains calcite. However, the white-colored plaster from the chamber exhibits the presence of kaolinite in addition to calcite. The kaolinite was identified by its typical IR bands [77]: around 3600 cm$^{-1}$ due to the OH stretchings, and a series of bands at 1034, 1008, and 797 cm$^{-1}$ corresponding to Si-O stretching, Si-O-Al and Si-O-Si deformation vibrations of the aluminosilicates. In accordance with the XRD analysis, gypsum was identified in the white-colored plaster from the chamber based on the IR bands at 3533, 3405 and 1109 cm$^{-1}$ [78,79].

A similar result was observed with the spectra of the two powder samples of black-colored plasters. The analysis of the spectra confirmed the presence of calcite as a major component in both samples, but the characteristic bands of kaolinite are observed in the spectrum of the black samples from the chamber while not present in the spectrum of the sample from the dromos (Figure S18).

Unlike the ATR spectra of the white and black colored plasters, the spectra of the samples from the red plasters (red, dromos and red, chamber) are almost completely identical (Figure S19). This fact emphasizes that the same red pigments were used in dromos and the chamber. In addition to the calcite, both spectra showed absorbance bands in positions that can be attributed to natural pigment red ochre [80,81].

Keeping in mind that the hematite is absorbing in the far-IR region, the spectra of the powder samples of red layers were measured in CsI pellet in an extended spectral range (4000–200 cm$^{-1}$) and compared to reference red ochre, hematite and calcite (Figure 11). A good correspondence was observed with the spectrum of reference red ochre (Kremer pigments, 40020) in the interval 1200–800 cm$^{-1}$ and above 3200 cm$^{-1}$, although the spectrum

was again dominated by the absorbance of the calcite. Hematite is characterized by strong IR bands at 532, 455 and 322 cm$^{-1}$ [82]. The examined red layers showed IR bands at close positions, but this should be considered with caution as calcite and kaolinite also absorb at similar wavenumbers.

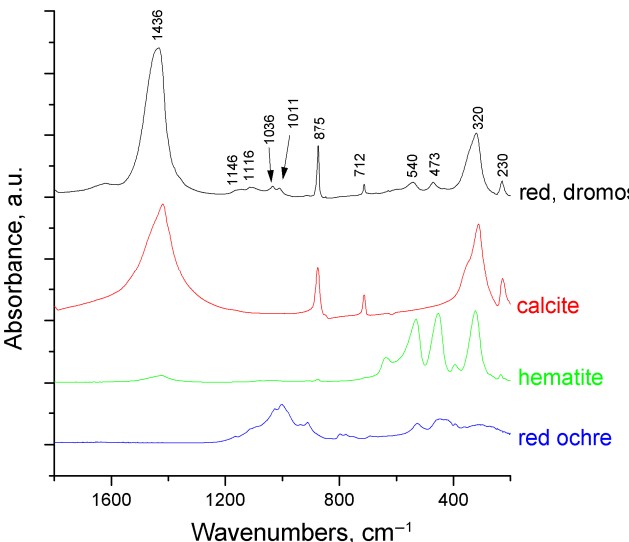

**Figure 11.** FTIR spectrum of powder sample collected from the red colored surface layer of domos, measured in CsI pellet, compared with reference red ochre, hematite and calcite.

The ATR-FTIR spectrum of the powder samples of the grey-colored layer from dromos (Figure S20) showed strong absorptions for calcite and, additionally, a band at 1028 cm$^{-1}$, which might be attributed to the stretching vibrations of apatite [83] if considered that IR bands around 3600 cm$^{-1}$ for the OH stretching of silicates are missing in this case. In contrast, in the IR spectrum of the powder samples of the brown paint layer, bands around 3600 cm$^{-1}$ and between 900 and 1220 cm$^{-1}$ are visible as a result of silicate mineral presence such as kaolinite or muscovite [84].

### 4.5. DSC Analysis

Considering that the detection of organic matter could be significantly hampered by the strong absorptions of inorganic materials, such as calcite, in the IR spectra, further research into the possible presence of organic binder was conducted by TG-DSC analysis of some samples. The thermal analysis measurements (DSC and TG) of the powder samples from red paint layers are depicted in Figure 12.

The DSC curves showed only one well-defined endothermic peak with a maximum of about 116 °C and 119 °C for both studied samples. It could be assigned to evaporable water, absorbed in the clayey materials, and is in good agreement with the slight weight loss registered for these samples up to 150 °C (Figure 12). The position of the maxima may slightly shift depending on the proportions of the clayey minerals in the material composition [85]. Besides, the presence of this endothermic peak allows discrimination of the red ochre from other red pigments, such as red bolus and pure hematite, where no such low-temperature effect is observed [86]. Therefore, in the present case too, it could be concluded that the red pigment in the colored plaster is natural red ochre. At the same time, the slight weight loss registered above 150 °C (about 2%–2.5%) could be attributed to not well-defined endothermic processes due to the dehydration of both gypsum and/or kaolinite that are present in these samples [87,88].

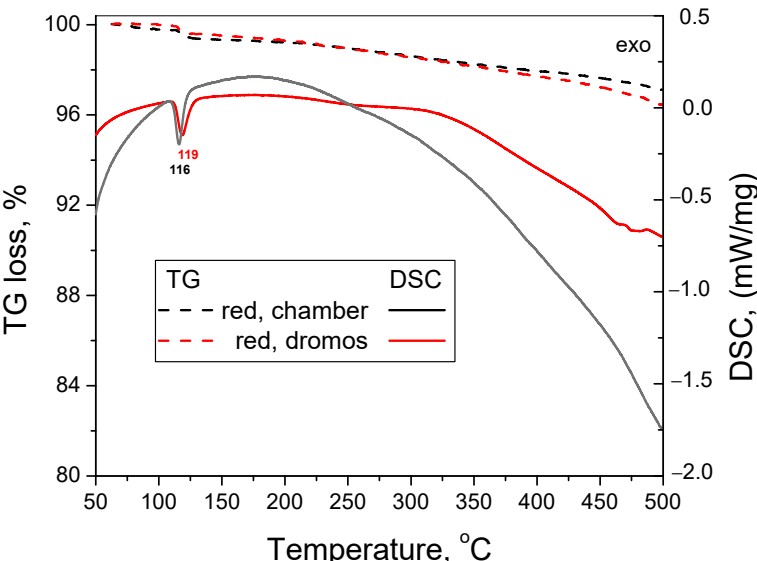

**Figure 12.** Thermal analysis measurements (DSC and TG) of the powder samples from red paint layers.

## 5. Discussion

According to historical evidence from ancient Greece and Rome, including the treatises of the Greek philosopher Theophrastus (371–287 BC), the Roman architect Vitruvius (80–15 BC), and the Roman scholar Pliny the Elder (23–79 AD) on plastering and coloring techniques, mortar was one of the main building materials used. It was applied either as a soldering substance between building elements (i.e., stone blocks, bricks) or as a plaster on the walls of the respective buildings. Different natural raw materials of local origin (such as limestone, dolomite, marble for the preparation of lime; sand from rock sources of different mineral composition, rock-forming, clay and micaceous minerals, etc.) were used to produce the mortar mixture, and the quality of the mortar was therefore strictly dependent on the ancient craftsmen's knowledge of the properties of minerals [89,90]. According to tradition, the decoration and painting of the walls was carried out by pre-treating them with successive application of several layers of mortar. Pigment was applied on top of the upper fine layer of mortar [91].

The results from the analytical study of the painting materials in the Thracian tomb near the village of Alexandrovo reveal the creative freedom of the ancient artist in applying wall decoration techniques. According to the analyzed data, in the case of the white (from the chamber), black, brown and grey decorations, the painting layer was applied on a two-layer lime plaster: one coarser layer between 0.5 and 0.8 cm thick and one finer layer between 0.15 and 0.30 cm thick. In the red decoration, three layers are clearly distinguishable: a rough layer (i.e., a lining plaster) about 0.8 cm thick; a white primer about 0.2 cm thick; and a light-colored fine layer less than 0.1 cm thick. The red wall decoration was carried out by applying pigments on top of a lightly colored pink primer.

The application of multiple layers of mortar is a crucial technological process that ensures even hardening of the plaster and determines its long-term stability. This process is a consequence of the properties of the slaked lime, as well as of the proportions between the composite materials in the mortar. The mineral composition of the fine mortar is a mixture of mainly quartz sand and calcite (as binder).

The painting layers' thickness, the presence of a large amount of calcium carbonate in all colored samples, and the absence of organic matter can be interpreted as evidence of the use of the fresco technique. This technique involves applying a water suspension of the pigment on wet, undried fine plaster. It was one of the most widely used in antiquity to paint large wall surfaces [92,93].

The pigment analysis showed that mostly natural pigments were used, including white—calcite and kaolinite; black pigment (amorphous C) of plant origin; red—natural

ochres (colored earths), grey—amorphous C, possibly of mixed plant and bone material; and brown—mixture of brown-colored earths with amorphous C.

The identified components are summarized in Table 2 along with the analytical techniques supporting their presence.

**Table 2.** The identified components in studied samples from the dromos and chamber and the analytical techniques supporting their presence.

| Color | Identified Components | LIBS | EPR | XRD | IR |
|---|---|---|---|---|---|
| **Dromos** | | | | | |
| Red | Calcite | + | + | + | + |
| | Hematite | + | | + | + |
| | Kaolinite | + | | + | + |
| | Quartz | + | | + | |
| | Gypsum | | | + | + |
| Black | Calcite | + | + | + | + |
| | Gypsum | | | + | + |
| | Muscovite | + | | + | |
| | Carbon | | + | | |
| | Dolomite | + | + | + | |
| White | Calcite | + | + | + | + |
| | Dolomite | + | + | + | |
| Grey | Calcite | + | + | + | + |
| | Apatite | | | + | + |
| | Carbon | | + | | |
| | Quartz | + | | + | |
| Brown | Calcite | + | + | + | + |
| | Muscovite | + | | + | |
| | Albite | + | | + | |
| | Carbon | | + | | |
| **Chamber** | | | | | |
| Red | Calcite | + | + | + | + |
| | Hematite | + | | + | + |
| | Kaolinite | + | | + | + |
| | Aragonite | + | | + | |
| White | Kaolinite | + | | + | + |
| | Gypsum | | | + | + |
| | Quartz | + | | + | |
| Black | Graphite | | + | + | |
| | Muscovite | + | | + | |
| | Kaolinite | + | | + | + |
| | Gypsum | + | | + | + |

The abundance in nature, the wide variety of colors and shades, and excellent durability made the application of natural ochres as pigments in wall painting very common [80,81]. They are natural mineral mixtures containing silicates and clay minerals mixed with iron oxides—goethite, limonite and hematite. LIBS, XRD and IR analysis showed the red layers contain aluminosilicate material (kaolinite) and hematite as the materials producing the red color. The brown layers were found to contain aluminosilicate material, specifically muscovite, in addition to calcite, as indicated by LIBS and XRD analysis. The FTIR spectrum analysis also showed characteristic bands of aluminosilicate and calcite, with no contribution from any known brown pigments. The EPR analysis indicated the presence of carbon. Therefore, it is likely that the brown color in this case is produced by a mixture of brown-colored earths, such as muscovite with carbon.

## 6. Conclusions

For the first time, the mural paintings from one of the most significant Thracian tombs on Bulgarian territory situated near the village of Alexandrovo, Bulgaria, dating from the fourth century BC, were studied using complementary analytical methods. Through the application of Laser-Induced Breakdown Spectroscopy (LIBS), X-ray Powder Diffraction (XRD), Paramagnetic Electron Resonance Spectroscopy (ESR), Attenuated Total Reflectance Fourier Transform Infrared Spectroscopy (ATR-FTIR) and thermal analysis measurements (DSC and TG), a comprehensive understanding of the techniques and materials used was achieved.

The results demonstrated that the ancient Thracian artists were well-skilled in the creation of mural decoration based on the properties of lime, which they applied successively in several layers to achieve better carbonization, on the one hand, and uniform coloring with the pigment, on the other hand. The pigmented layer was applied on top of a fine mortar. The presence of calcite in all pigmented layers and the lack of organic binders indicated *fresco* as the main painting technique. The pigments used in the wall decoration are natural minerals suitable for the fresco technique because of their good chemical resistance. The color shades were achieved by using various pigment mixtures.

The current results contribute to the knowledge, research and study of monumental decoration and painting techniques acquired and developed in ancient times. They will allow the study of the similarities and distinctions of the Thracian tombs found on the territory of Bulgaria, as well as the Balkan region.

**Supplementary Materials:** The following supporting information can be downloaded at: https://www.mdpi.com/article/10.3390/min14040374/s1, Table S1. The spectral lines used for element detection in the LIBS analysis; Table S2. EPR parameters of reference black pigments and ashed materials; Table S3. Identified components and IR characteristic bands of the studied samples, measured by ATR-FTIR technique; Figures S1–S8 Photos of the studied fragment with respective scales; Figure S9. Two sections of LIBS spectra in the spectral region (a) 300 nm to 303 nm and (b) 394 nm to 395 nm obtained from fragments red, chamber; Figures S10–S17. Diffraction pattern of the studied samples; Figure S18. ATR-FTIR spectra of the black colored surface layers from dromos and chamber; and reference materials calcite, gypsum and kaolinite; Figure S19. ATR-FTIR spectra of the red-colored surface layers from dromos and chamber and reference calcite, gypsum and kaolinite; Figure S20. ATR-FTIR spectra of the dark grey colored surface layer from dromos and reference material hydroxyapatite.

**Author Contributions:** Conceptualization, G.N., R.S., B.S., G.A. and V.T.; formal analysis, G.N., R.S., B.S., G.A. and V.T.; funding acquisition, R.S. and G.N.; investigation, G.A., R.K., V.T., V.M., D.Y., M.D., G.N., R.S. and B.S.; methodology, G.A., V.M., G.N., R.S. and B.S.; project administration, G.N., R.S., B.S., G.A. and V.T.; supervision, G.N., R.S., B.S., G.A. and V.T.; validation, G.A., R.K., V.T., V.M., D.Y., M.D., G.N., R.S. and B.S.; visualization, G.A., R.K., V.T., V.M., D.Y., M.D., G.N., R.S. and B.S.; writing—original draft, G.A., R.K., V.T., V.M., D.Y., M.D., G.N., R.S. and B.S.; writing—review and editing, G.A., R.K., V.T., V.M., D.Y., M.D., G.N., R.S. and B.S. All authors have read and agreed to the published version of the manuscript.

**Funding:** This research was funded by Bulgarian Academy of Sciences under project "The Thracians—genesis and development of the ethnos, cultural identities".

**Data Availability Statement:** The data presented in this study are available upon request from the corresponding author.

**Acknowledgments:** Equipment of INFRAMAT (Research Infrastructure from National roadmap of Bulgaria), supported by Contract Д01-306/2021 with Bulgarian Ministry of Education and Science is used in the present investigations.

**Conflicts of Interest:** The authors declare no conflicts of interest.

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
