# Peer review of "Multi-Analytical Analysis of Decorative Color Plasters from the Thracian Tomb near Alexandrovo, Bulgaria"

_minerals, doi:10.3390/min14040374_

Round 1

Reviewer 1 Report

Comments and Suggestions for Authors

This article "Multi-analytical analysis of decorative colored plasters from the Thracian tomb near Alexandrovo, Bulgaria" seems to be only a study case, it is not suitable for a scientific mineral journal. 

The introduction part talked about the historty of this tombe, but it should give a novety of this paper in scientific aspect. 

In the materials and methods part, the photographs of the samples are missing their scales, and the micro samples are not presented in the paper, including the states and the scales.

Each technique used is not well described the purpose of utilization and the implementation.

In the results part, the spot size of LIBS measurement varies, the authors should avoid it. It leads to the comparison of intensity useless.

Figure 6 is unreadable, and the intensity of different samples cannot be compared because of the different ablation.

The abbreviation "EPR" has not been defined in this paper. Figure 7 the right figures missed the points.

Figure 9, the number of the sample was not given.

All figures in section 3.4 were missing the unit of Y coordinate. 

The DSC analysis did not provide any additional information. It is not necessary to use it. Because before, the authors already confirmed no presence of organic materials.

I suggest the authors to summarize the information obtained from the techniques used according to the samples in a table.

Author Response

Dear Review,

Thank you for your comments and suggestions on the completeness of our manuscript.

All the revisions in the revised manuscript are highlighted with a yellow background.

Detailed corrections are listed below point by point:

In the materials and methods part, the photographs of the samples are missing their scales, and the micro samples are not presented in the paper, including the states and the scales.

 Response: The photographs of the fragments with respective scales were added in the Supplementary Materials (Figures S1-S8).

Each technique used is not well described the purpose of utilization and the implementation.

Response:

An additional section “Analytical techniques applied in the study and related works” was added.

In the results part, the spot size of LIBS measurement varies, the authors should avoid it. It leads to the comparison of intensity useless.

Response: When we wrote that the spot size varies, we meant that generally in the LIBS measurements it varies. In the current study the spot size did not vary significantly because the analysis is performed at the same laser pulse energy and same number of pulses at single spot. To avoid this confusion and to be more clear the sentence has been removed and the following text has been added:
“Although there are small differences in ablation from sample to sample the ratio of the intensities of the spectral lines is indicative of the relative abundance of the elements. In order to be able to compare the intensities of the spectral lines of the elements, the experimentally measured spectra of the samples were normalized by the method.”

Figure 6 is unreadable, and the intensity of different samples cannot be compared because of the different ablation.

Response: In order to compare correctly the content of calcium and iron in the different samples, the measured spectra were normalized according to the method. This has been added in the caption below the edited graph.

The abbreviation "EPR" has not been defined in this paper. Figure 7 the right figures missed the points.

Response: The “missing points” are due to the specific structure of some of the studied fragments. Sample 5 white, dromos consists of only fine and coarse mortar as no pigmented surface could be seen, while sample brown, dromos doesn’t have coarse mortar.

These peculiarities are explained in the text – Two samples deviate from the general appearance. The white sample from the dromos, represents a one-layer slab with smoother upper and rougher lower surfaces. The brown sample from the dromos consists of only fine mortar with pigmented upper part.

Figure 9, the number of the sample was not given.

Response: The names of the samples were corrected in Figure 9.

All figures in section 3.4 were missing the unit of Y coordinate.

Response: Intensity of the IR spectra are given in arbitrary units (a.u.), often used in ATR-FTIR spectra.

The DSC analysis did not provide any additional information. It is not necessary to use it. Because before, the authors already confirmed no presence of organic materials.

Response:

Having in mind that detection of organic matter could be significantly hampered by the strong absorptions of inorganic materials such calcite in the IR spectra, it was considered more reliable to use a complementary method to research into the possible presence of organic binder such as TG-DSC analysis. In a similar situation, recently the method was successfully applied for the identification of beeswax in the mural decoration of Thracian monuments.

I suggest the authors to summarize the information obtained from the techniques used according to the samples in a table.

Response: Table 2 with the information obtained from the techniques used were added.

Reviewer 2 Report

Comments and Suggestions for Authors

The manuscript presents the novelty of a complex multianalytical analysis without precedence the pigments and pigmented plasters from Thracian tombs.

The importance of the topic is clearly presented, the technical approach and exposition of the results is sound and convincing.

In order to further enhance the quality of the paper, I suggest some minor changes on its structure:

-An additional section of "related work" between the introduction and the materials and methods where you introduce references to the used analytic techniques in similar contextes and explain why they are useful and chosen

-An additional final section of "conclusions" where you do a brief sum-up of your work, deliver your view over the whole work and highlighting its importance, and possible future lines of work after the drawn results

Author Response

Dear Review,

Thank you for your suggestions on the completeness of our manuscript.

In revised manuscript were added two new sections:

2. Analytical techniques applied in the study and related works

6. Conclusions

The new texts in the revised manuscript are specified with a yellow background.

Reviewer 3 Report

Comments and Suggestions for Authors

Dear authors,

This manuscript is well written and may be of interest to the Journal Minerals readership but may be accepted for publication only after minor revision. Here are my comments.

Comment 1: English correction required.

Comment 2: Lines 36-37 …,,The Thracian civilization reached its peak between the 5th and 3rd centuries BC in present-day Bulgaria.” - Please insert a reference after such a claim in the article.

Comment 3: Lines 56-57 …,,Figure 1. Plan 1:50 (a) and section 1:50 (b) of the tomb at the village of Alexandrovo (author M. Kamenova).” - Please enter the year when the author M. Kamenova created this plan.

Comment 4: Lines 74-75 …,,Figure 2. Part of the funeral feast scene from the chamber of the tomb at the village of Alexandrovo 74 (author A. Cosentino).” - Please enter the year when the author A. Cosentino created this picture.

Comment 5: Line 81 …,, an inscription in Greek “ΚΟΞΙΜΑСΗС ΧΡΗСТОС” is scratched” – I suggest removing the word Greek, leaving out whose script it is.

Comment 6: Lines 90-91 …,, Figure 3. Part of the frieze with hunting scenes from the chamber of the tomb at the village of the 90 Alexandrovo (author A. Cosentino).” - Please enter the year when the author A. Cosentino created this picture.

Comment 7: Lines 120-124 – …,,The experimental techniques used in the current study are Laser-Induced Breakdown Spectroscopy (LIBS), X-Ray Powder Diffraction (XRD), Paramagnetic Electron Resonance Spectroscopy (ESR), Attenuated Total Reflectance Fourier Transform Infrared Spectroscopy (ATR-FTIR) and thermal analysis measurements (DSC and TG).” -Please remove this sentence from this section of the article and move it to the Materials and Methods section.

Comment 8: Please enter the number of cards for the mineral species you have identified in the article. It's up to you to choose whether to drag them into line 152 or the section from line 314.

Comment 9: Lines 387-392 - Please enter the appropriate reference after each IR band listed.

Comments on the Quality of English Language

English correction required.

Author Response

Dear Review,

Thank you for your comments and suggestions on the completeness of our manuscript.

All the revisions in the revised manuscript are specified with a yellow background.

Detailed corrections are listed below point by point:

Comment 1: English correction required.

Response:

The manuscript was revised and all found grammatical errors were removed. Also the text was stylistically improved – the revised sentences are highlighted in yellow.

Comment 2: Lines 36-37 …,,The Thracian civilization reached its peak between the 5th and 3rd centuries BC in present-day Bulgaria.” - Please insert a reference after such a claim in the article.

A new reference was added according to the suggestion:

Stoyanova. D. Tomb Architecture. In A Companion to Ancient Thrace. Valeva, J., Nankov, E., Graninger, D. Eds.; Wiley-Blackwell, 2015, 158-179.

 Comment 3: Lines 56-57 …,,Figure 1. Plan 1:50 (a) and section 1:50 (b) of the tomb at the village of Alexandrovo (author M. Kamenova).” - Please enter the year when the author M. Kamenova created this plan.

Response:

The year was added to the caption of Figure 1.

Comment 4: Lines 74-75 …,,Figure 2. Part of the funeral feast scene from the chamber of the tomb at the village of Alexandrovo 74 (author A. Cosentino).” - Please enter the year when the author A. Cosentino created this picture.

Response: The year was added to the caption of Figure 2.

Comment 5: Line 81 …,, an inscription in Greek “ΚΟΞΙΜΑСΗС ΧΡΗСТОС” is scratched” – I suggest removing the word Greek, leaving out whose script it is.

Response:

The word Greek was removed.

Comment 6: Lines 90-91 …,, Figure 3. Part of the frieze with hunting scenes from the chamber of the tomb at the village of the 90 Alexandrovo (author A. Cosentino).” - Please enter the year when the author A. Cosentino created this picture.

Response: The year was added to the caption of Figure 3.

Comment 7: Lines 120-124 – …,,The experimental techniques used in the current study are Laser-Induced Breakdown Spectroscopy (LIBS), X-Ray Powder Diffraction (XRD), Paramagnetic Electron Resonance Spectroscopy (ESR), Attenuated Total Reflectance Fourier Transform Infrared Spectroscopy (ATR-FTIR) and thermal analysis measurements (DSC and TG).” -Please remove this sentence from this section of the article and move it to the Materials and Methods section.

Response: This section was moved to section Materials and Methods.

Comment 8: Please enter the number of cards for the mineral species you have identified in the article. It's up to you to choose whether to drag them into line 152 or the section from line 314.

 Response: The numbers of cards for the mineral species were added.

Comment 9: Lines 387-392 - Please enter the appropriate reference after each IR band listed.

Response: Appropriate references for the characteristic IR bands and assignment were added according to the suggestion.

Round 2

Reviewer 1 Report

Comments and Suggestions for Authors

The authors have answered most of the questions, but for the IR spectra even if in arbitrary units (a.u.), it should be have a direction of increasing.

Author Response

Dear Review,

The units of IR intensity in Figure 10 were added.